# *When LLMs Meet Cunning Texts:* A Fallacy Understanding Benchmark for Large Language Models

**Yinghui Li**[1,*] **Qingyu Zhou**[2,*,†] **Yuanzhen Luo**[*], **Shirong Ma**[1],
**Yangning Li**[1]**, Hai-Tao Zheng**[1,†]**, Xuming Hu**[3,†]**, Philip S. Yu**[4]
[1]Tsinghua University, [2] Bytedance Inc.
[3]The Hong Kong University of Science and Technology (Guangzhou)
[4]University of Illinois Chicago
liyinghu20@mails.tsinghua.edu.cn

## Abstract

Recently, Large Language Models (LLMs) make remarkable evolutions in language understanding and generation. Following this, various benchmarks for measuring all kinds of capabilities of LLMs have sprung up. In this paper, we challenge the reasoning and understanding abilities of LLMs by proposing a **F**a**L**lacy **U**nderstanding **B**enchmark (**FLUB**) containing cunning texts that are easy for humans to understand but difficult for models to grasp. Specifically, the cunning texts that FLUB focuses on mainly consist of the tricky, humorous, and misleading texts collected from the real internet environment. And we design three tasks with increasing difficulty in the FLUB benchmark to evaluate the fallacy understanding ability of LLMs. Based on FLUB, we investigate the performance of multiple representative and advanced LLMs, reflecting our FLUB is challenging and worthy of more future study. Interesting discoveries and valuable insights are achieved in our extensive experiments and detailed analyses. We hope that our benchmark can encourage the community to improve LLMs' ability to understand fallacies. Our data and codes are available at https://github.com/THUKElab/FLUB.

## 1 Introduction

Large Language Models (LLMs) have shown great abilities to understand human languages, including information extraction [1], text correction [2], humor understanding [3], etc. Researchers have constructed numerous benchmarks to evaluate LLMs in various aspects [4–8]. By using constructed benchmarks to interact with LLMs, researchers can analyze the behavior of LLMs to compare the performance of different LLMs and study how to further improve LLMs in a targeted manner.

Although many LLM benchmarks have sprung up, we believe that existing benchmarks are not challenging enough to truly measure the human-like intelligence of LLMs. In particular, we are still wondering whether LLMs can understand cunning texts that may contain misleading, wrong premise, intentional ambiguity, and so forth, considering that almost all LLMs are trained on "cleaned" and "correct" corpora. Therefore, we build a **F**a**L**lacy **U**nderstanding **B**enchmark (**FLUB**) to challenge LLMs for solving these problems.

Figure 1a shows the running examples from FLUB. From these cases, we directly feel the different behaviors of LLMs and humans when facing cunning texts. In the first example, LLMs ignore the

---

[*]indicates equal contribution.
[†]Corresponding authors

38th Conference on Neural Information Processing Systems (NeurIPS 2024) Track on Datasets and Benchmarks.

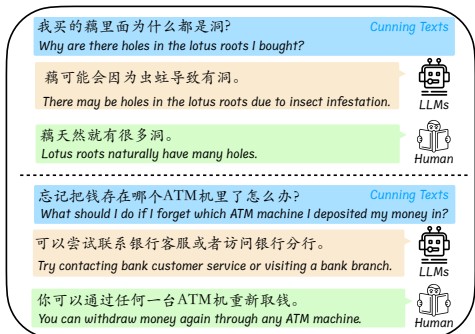

(a) The examples of how LLMs and humans perform when faced with cunning texts. The LLM we use is ChatGPT-3.5 on Jan 23, 2024.

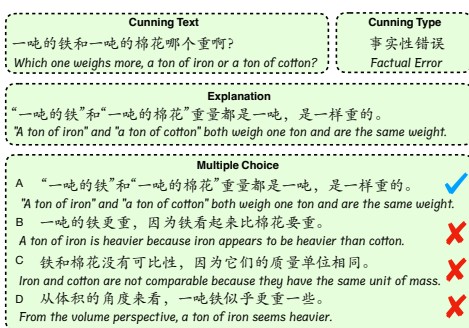

(b) We design three tasks, namely Cunning Type Classification, Fallacy Explanation, and Answer Selection (i.e., Multiple Choice).

Figure 1: The running examples and annotation examples of FLUB.

common sense that the lotus root itself has many holes in its structure and fall into the trap of the cunning text, wrongly judging that the holes in the lotus root are caused by insect infestation. In the second example, LLMs fail to see the logic that depositing money into random ATMs does not create problems and therefore give an answer that seems reasonable but is absurdly laughable. In fact, these cunning texts for LLMs are very easy to handle for human intelligence. **Therefore, it is very urgent and meaningful to construct a benchmark composed of cunning texts to evaluate and thereby promote the improvement of LLMs' fallacy understanding capabilities.**

Inspired by the above motivation, we collect real cunning texts as our raw data from a famous Chinese online forum, the "Ruozhiba" (retard forum)[3]. This forum is popular for its cunning and unreasonable posts, which are generally easy for humans to understand but challenging for LLMs. The characteristics of the posts contained in this forum are consistent with our research motivation, so choosing it as the data source well supports FLUB's evaluation of LLMs' fallacy understanding ability. After data cleaning and annotating of cunning types, FLUB has 8 fine-grained types of cunning texts and most of the texts in FLUB fall into two types of fallacy, namely, faulty reasoning and word game. Moreover, we also manually annotated one correct answer (i.e., the explanation of the cunning text) and three confusing wrong answers for each input text in FLUB, as shown in Figure 1b.

Based on our constructed FLUB and its annotation information, we design three tasks with increasing difficulty to test whether the LLMs can understand the fallacy and solve the "cunning" texts. Specifically, (1) **Answer Selection**: The model is asked to select the correct one from the four answers provided by FLUB for each input text. (2) **Cunning Type Classification**: Given a cunning text as input, the model is expected to directly identify its fallacy type defined in our scheme. (3) **Fallacy Explanation**: We hope the model sees a cunning text and intelligently generates a correct explanation for the fallacy contained in the text, just like humans, without falling into its trap.

In our experiments, we select representative and advanced LLMs to be evaluated on FLUB. Our empirical study reveals: (1) LLMs are very poor in their ability to perceive fallacy types in cunning texts. (2) For a specific task, LLMs with larger parameter sizes do not always perform better. (3) There is a close relationship between the Answer Selection task and the Fallacy Explanation task, and the interaction between them is critical to promoting the understanding of fallacies in LLMs. (4) On FLUB, the widely used Chain-of-Thought and In-context Learning techniques deserve further improvement and research. We believe that our proposed FLUB and all our findings are crucial for LLMs to comprehend the fallacy and handle cunning texts in the real world.

---

[3]https://tieba.baidu.com/f?kw=%E5%BC%B1%E6%99%BA&ie=utf-8

| Cunning Type | Definition | Example |
|---|---|---|
| 错误类比
False Analogy | 由于事件A的发生具有或伴随有某种属性，从而错误地类比出与事件A相似的事件B也应该具有该属性，或者错误地类比出与事件A相反的事件B应该具有相反的属性。
Due to the occurrence of event A having or being accompanied by a certain attribute, it is erroneously analogized that event B, which is similar to event A, should also have that attribute, or that event B, which is opposite to event A, should have the opposite attribute. | 很多人出门后担心刚刚没有关门，为什么进门后不担心刚刚没有开门？
Many people worry about forgetting to close the door when they leave home. Why don't they worry about whether they have opened the door when they come in? |
| 冷笑话
Lame Jokes | 由于缺乏对某个常识或事实的认知，从而得出某个不符合逻辑的问题或结论。注意，该句子往往因为其不寻常的认知缺失从而导致该句子可能令人发笑。
Due to a lack of understanding of a common sense or fact, a illogical question or conclusion can be drawn. Note that this sentence may be funny due to its unusual cognitive impairment. | 忘记把钱存在哪个ATM机里了怎么办？
银行好几台ATM机，还长得都一样。
What should I do if I forget which ATM I deposited money into? The bank has several ATMs, and they all look the same. |
| 字音错误
Phonetic Error | 通过改变固定词汇中多音字的发音，从而得到新的句子。注意，如果读者没有察觉到句子中发音的改变，会导致读者认为该句子不符合逻辑。
Sentences obtained by changing the pronunciation of polyphonic words in fixed vocabulary. Note that if the reader does not appreciate the change in pronunciation in the sentence, it will lead the reader to think that the sentence is illogical. | 因为美国队长，小明每次在美国排队都要排一个多小时。
Because of Captain America (also read as "long queues in America" in Chinese), XiaoMing has to wait over an hour whenever he queue in the U.S. |
| 歧义
Ambiguity | 通过改变句子中某个多义词的词义，从而得出不符合逻辑的问题或结论。
By changing the meaning of a polysemy word in a sentence, illogical questions or conclusions can be drawn. | 语文老师说我的句子是病句，我应该给这个病句吃头孢，还是打点滴呢？
My teacher said the sentence is grammatically wrong. Should I give this sentence some antibiotics or administer an IV drip? |
| 悖论
Paradox | 句子或者问题的表述前后矛盾。
The expression of a sentence or question is contradictory. | "凡事无绝对"这句话过于绝对。
The phrase "Nothing is absolute" is too absolute. |
| 事实性错误
Factual Error | 由于缺乏对某个事实的认知，或者对事实进行扭曲，从而提出无意义的问题或结论。
Due to a lack of understanding or distortion of a fact, meaningless questions or conclusions are raised. | 一吨的铁和一吨的棉花哪个重啊？
Which one weighs more, a ton of iron or a ton of cotton? |
| 推理错误
Reasoning Error | 从一个事件中推断出一个错误或者无意义的结论，或者颠倒了事件的因果关系。
Inferring an incorrect or meaningless conclusion from an event, or reversing the causal relationship of the event. | 根据我在养老院的调查数据，我国的人口老龄化已经相当严重了。
According to my survey data from nursing homes, the aging of the population in our country has become quite severe. |
| 文字游戏
Word Game | 错误地改变句子中文字的意思或含义，在此基础上提出问题或者得出结论。
Mistakenly changing the meaning of words in a sentence, and based on this, raising questions or drawing conclusions. | 人类70%是水，所以10个人里有7个人是水伪装成的人！
70% of the human body is water, so 7 out of 10 people are water disguised as humans! |
| 未分类 Undefined | 句子本身具有错误，或者句子的表述不符合正常逻辑，但是不属于上述任何一个类别。
The sentence itself has errors, or the expression of the sentence does not conform to normal logic, but does not belong to any of the above categories. | 在高速路的服务区开酒吧有可行性吗？
Is it feasible to open a bar at a highway service area? |

Figure 2: The definitions and examples of the cunning types in FLUB.

## 2 The FLUB Benchmark

### 2.1 Benchmark Construction

**Data Collection**   We collect raw text data from "Ruozhiba" in Baidu Tieba [4]. "Ruozhiba" is one of the most famous online forums in the Chinese internet community, and people often post interesting or "silly" texts on it just for fun. In addition, the recent study [9] also shows that the Ruozhiba data is very useful for improving the ability of Chinese LLMs. We find that many of the posts on this forum are tricky texts or brain-teaser-like texts, which is exactly in line with our purpose of using cunning texts to challenge LLMs, so we utilize this forum as our data source. As a result of automatic crawling, we initially collect 9,927 candidate posts. Notably, according to the Baidu Bar agreement [5], the data on Baidu Tieba can be used for academic research free of charge and without liability.

**Data Cleaning**   We employ annotators to manually filter out irrelevant posts that do not present cunning texts. Since the collected original posts contain irrelevant content such as links and images, we also require annotators to extract the fallacious and illogical contents from the raw post and rewrite them into a complete sentence. Besides, it is worth noting that we carefully ensure that the texts in FLUB are ethical texts. This process includes user information anonymization, sensitive information removal, and filtering of impolite posts. In total, we obtain 834 data samples to form FLUB.

**Data Annotation**   To ensure the annotation quality, our criteria for selecting annotators is that the person must be a native Chinese speaker and have a bachelor's degree. In addition, because FLUB comes from the online forum, we also require annotators to have more than five years of experience as netizens. The detailed annotation workflows are as follows:

1. **Cunning Type Annotation**: We first define 8 cunning types within the collected texts along with their corresponding examples, as shown in Figure 2. Specifically, our core authors make a comprehensive summary based on careful observation of the 9,927 initial candidate posts, thus defining 8 types. Subsequently, each data sample is processed by three junior annotators, who are required to select an appropriate cunning type for the sample. We achieve the initial annotation results based on the voting results among three annotators. The initial annotation results become the final annotation information after being reviewed by the senior annotator (and modified if necessary). Particularly, there are still a small number

---
[4] https://tieba.baidu.com
[5] https://baike.baidu.com/item/%E8%B4%B4%E5%90%A7%E5%8D%8F%E8%AE%AE/8397765

of samples that fall into multiple types. For these samples, senior annotators and our core authors will discuss carefully and select the main type (i.e., the most obvious type among multiple types) as the annotation result.

2. **Correct Explanation Annotation**: We assign two junior annotators to write the explanation or answer for each sample independently. We ask them to try to explain the given text in a detailed, objective, and unambiguous way. The senior annotator then selects (and modifies if necessary) the more suitable text written by the two junior annotators.

3. **Wrong Candidates Annotation**: This part annotation is to obtain the wrong candidate answer that may be likely to be answered incorrectly for each input text. We assign three junior annotators for each sample and require each of them to write three different incorrect answers based on their understanding of the text. Particularly, we emphasize to each junior annotator that the three different wrong answers they write should ensure diversity and resemble as much as possible the answers that LLMs can easily produce. For each sample's nine initial incorrect answers, the senior annotator selects the three most challenging sentences as the final wrong candidates.

Since the annotation difficulty of different information is different, the salary we pay to the annotators we employ is also different. Specifically, we pay each person who annotates the cunning type $0.5 per sample, each person who writes the correct explanation $1 per sample, and each person who writes the wrong candidates $2 per sample. In addition to the junior annotators providing the initial annotation results, we also set three senior annotators with a salary of $2 per sample, who are responsible for carefully checking the correctness of the annotation results provided by the junior annotators.

It is worth mentioning that we have prepared sufficient and representative samples for annotators to learn and pre-annotate to ensure that they fully understand the information we want to annotate before they officially start annotation. Specifically, we select senior annotators based on their performance in the pre-annotation process. If an annotator's success rate is above 95%, he or she will be appointed as a senior annotator. In addition, it is worth mentioning that, all of our formal annotators have a success rate of over 80% in the pre-annotation process. At the same time, to avoid bias caused by the subjectivity of annotators as much as possible, our core authors also carefully checked the final annotation results of each data sample. Our entire annotation process lasted 2 weeks.

## 2.2 Dataset Analysis

**Data Size**    FLUB comprises 834 samples that span 8 cunning types. It is worth emphasizing that the data size is not directly related to the evaluation effectiveness of a LLM benchmark. For example, TruthfulQA [10] and FreshQA [11], these benchmarks that have been widely used and had deep impacts, only have 817 and 500 test samples respectively. The main reasons limiting the size of FLUB are that it is derived entirely from real-world online forum posts and our rigorous high-quality data cleaning process, which retained 834 final samples from 9,927 candidate posts.

**Data Distribution**    As for the cunning type distribution of FLUB, most data in FLUB belong to the types of reasoning errors (53.4%) and word games (28.7%). This is because these two types of posts appear widely in "Ruozhi Bar" forum whose purpose is to challenge human intelligence. A large number of cunning texts involving reasoning errors and word games ensure that FLUB is challenging enough. Besides, we observe that some types of texts are relatively rare, such as phonetic errors (0.6%). In fact, this is because our data come entirely from the real world and are all carefully constructed by netizens. Cases of cunning texts caused by phonetic errors are indeed rare in the real world. To eliminate the impact of type imbalance when FLUB evaluating LLMs, we choose the F-1 score as the evaluation metric which comprehensively considers the type coverage.

**Annotation Quality**    Since cunning type annotation is essentially a classification process performed by multiple annotators, we analyze the annotation quality of this information. Specifically, we calculate Fleiss' Kappa [12] to reflect the three junior annotator's Inter-Annotator Agreement (IAA). Our final obtained Fleiss Kappa result is greater than 0.767, which shows that our annotation results have excellent consistency and quality [13]. On the other hand, we further ensure annotation quality by checking the annotation and modification results of the senior annotators. According to our statistics, senior annotators modified a total of 159 initial annotation results of data samples, that is, the modification rate of senior annotators was 19.06%. This reflects the excellent workload of our

senior annotators and also reflects the high quality of our dataset. Moreover, after further checking of the modification results of the senior annotators by our core authors, we found that the main reason for the modifications was the disagreement between the senior annotators and the junior annotators on the cunning types (most of the cases were the ones we mentioned before that may fall into multiple types of samples). For these cases, our core authors made the most reasonable choices and personally modified the annotation results to maximize the quality of the annotation. After all, no one knows the full picture of our work better than our core authors.

## 2.3 Benchmark Task Setups

To evaluate the fallacy understanding ability of LLMs, we design three benchmark tasks on FLUB: Answer Selection, Cunning Type Classification, and Fallacy Explanation. For each task, we design prompts to guide LLMs on the expected output. We also explore the prompting strategies of Chain-of-Thought and In-context Learning to conduct in-depth exploration on FLUB. The details of our designed prompts are shown in Appendix A. Below we introduce the details of our three tasks:

**Task 1: Answer Selection**  In Task 1, LLMs are required to select the correct answer from four given candidate explanations for each input text. The annotation of candidate explanations is illustrated in Figure 1b. In general, each sample in this task is a tuple $\{p, q, O_A, O_B, O_C, O_D, l\}$, where $p$ is our given prompt as shown in Appendix A, $q$ is the input text, $O_A$, $O_B$, $O_C$, and $O_D$ are four candidate explanations, and $l \in \{A, B, C, D\}$ is the golden label indicating $O_l$ is the correct explanation. The design motivation of this task is to test whether LLMs can distinguish right from wrong when seeing the correct and wrong answers in the context of a given cunning text.

**Task 2: Cunning Type Classification**  If LLMs are directly tasked with determining the corresponding cunning type, it will help us in conducting an initial automated assessment of the LLM's understanding ability. The cunning type classification task is specifically designed to evaluate whether LLMs can classify the cunning text into categories aligned with human intuition based on the hidden irrational aspects within the current text. The annotated problem types are shown in Figure 2. During task evaluation, all the problem types will be combined with the prompt to allow LLMs to directly pick the correct type of cunning text.

**Task 3: Fallacy Explanation**  To further test whether LLMs truly understand the given cunning text, we design the explanation task. In this task, the designed prompt and input texts are directly input into LLMs, enabling them to "read" input texts and generate corresponding explanations. Note that since some texts are not expressed in the form of inquiries, we also set a prompt to guide LLMs in identifying the question (See Appendix A). The generated explanations will be compared with the correct explanation for evaluation. If LLMs can generate reasonable explanations, we believe that they have at least developed the ability to identify and avoid the traps of cunning texts.

**Automatic Evaluation Metrics**  For Task 1, we calculate **Accuracy** directly based on the LLMs' selection results. For Task 2, considering that there are a few cunning types in FLUB with small sample size, we choose the **F-1 Score** to measure the performance of LLMs because it focuses on both the accuracy of model prediction and the coverage for positive class samples, thereby effectively avoiding bias caused by type imbalance and ensuring the rationality and reliability of evaluation. To evaluate the quality of LLMs' generated explanations in Task 3, inspired by MT-Bench [14], we construct prompts that incorporate the task instruction, input texts, LLM's explanations, and reference answers. These prompts are fed into GPT-4, which is tasked with assigning a **GPT-4 Score** ranging from 1 to 10. The prompt for the automated evaluation is illustrated in Appendix B.

**Human Evaluation Settings**  For Task 1 and Task 2, **we conduct human evaluations to explore how well human-level intelligence could perform these two tasks.** To ensure the fairness of the comparison between humans and LLMs, we hire 3 new persons who do not participate in the construction process of FLUB. After briefly introducing them to the objectives of Task 1 and Task 2 (without introducing additional knowledge and information), let them directly carry out selection and classification. For the human evaluation of Task 3, **we mainly want to verify the effectiveness of the automatic GPT-4 score we use,** therefore, we hire 3 evaluation annotators to rate LLMs' explanations, with scores ranging from $\{1, 2, 3, 4, 5\}$. To ensure an accurate evaluation of the explanations of

Table 1: We **bold** the optimal and underline the suboptimal of closed/open-source models. We report the overall performance by calculating the **geometric mean** of the three tasks. We color the result that Chain-of-Thought (CoT) brings positive / negative gain as green↑ / red↓ .

| Models | Open Source | Selection Accuracy | | Classification F-1 Score | | Explanation GPT-4 Score | | Overall Performance | |
|---|---|---|---|---|---|---|---|---|---|
| | | w/o CoT | CoT | w/o CoT | CoT | w/o CoT | CoT | w/o CoT | CoT |
| ERNIE-Bot-3.5-Turbo [15] | ✗ | 32.97 | 34.65↑ | 1.99 | 6.09↑ | 5.78 | 5.83↑ | 7.24 | 10.72↑ |
| ERNIE-Bot-3.5 [15] | ✗ | 52.76 | 38.37↓ | 10.33 | 11.15↑ | 6.35 | 6.22↓ | 15.13 | 13.86↓ |
| ERNIE-Bot-4.0 [15] | ✗ | 75.66 | 71.34↓ | 11.84 | **14.42**↑ | 7.73 | 8.11↑ | 19.06 | 20.28↑ |
| GPT-3.5-Turbo [16] | ✗ | 50.48 | 48.08↓ | 3.09 | 6.15↑ | 6.23 | 7.00↑ | 9.91 | 12.74↑ |
| GPT-4-Turbo [16] | ✗ | **79.38** | **82.73**↑ | **12.31** | 13.97↑ | **8.95** | **9.21**↑ | **20.60** | **22.00**↑ |
| ChatGLM3-6B [17] | ✓ | 35.85 | 35.01↓ | 7.48 | 9.34↑ | 4.98 | 4.82↓ | 11.01 | 11.64↑ |
| Qwen-7B-Chat [18] | ✓ | 38.49 | 33.69↓ | 8.00 | 10.97↑ | 5.39 | 5.65↑ | 11.84 | 11.98↑ |
| Qwen-14B-Chat [18] | ✓ | 42.57 | 43.05↑ | **10.34** | 10.44↑ | 5.24 | 6.24↑ | 13.21 | 14.10↑ |
| Qwen-72B-Chat [18] | ✓ | **58.63** | **61.51**↑ | 9.32 | **12.26**↑ | **7.34** | **7.90**↑ | **15.89** | **18.13**↑ |
| Yi-6B-Chat [19] | ✓ | 32.37 | 29.26↓ | 8.87 | 9.84↑ | 5.73 | 5.39↓ | 11.81 | 11.58↓ |
| Yi-34B-Chat [19] | ✓ | 47.96 | 48.80↑ | 4.74 | 11.70↑ | 6.97 | 7.52↑ | 11.66 | 16.17↑ |
| Baichuan2-7B-Chat [20] | ✓ | 43.17 | 37.17↓ | 1.02 | 4.45↑ | 5.48 | 4.85↓ | 6.23 | 9.29↑ |
| Baichuan2-13B-Chat [20] | ✓ | 37.05 | 38.01↑ | 3.52 | 4.58↑ | 5.79 | 5.84↑ | 9.11 | 10.06↑ |
| Random | - | 25.00 | | 7.90 | | - | | - | |
| Human | - | 93.35 | | 63.69 | | - | | - | |

LLMs, we developed a set of scoring guidelines for annotators, including the definitions and relevant examples for each score. The scoring guidelines of human evaluation are presented in Appendix C.

When designing the GPT-4 scoring range and the human scoring range, we have different motivations. We hope that GPT-4's scoring range can be as unbiased and detailed as possible, so we set its scoring range to 1-10. But this scoring range is too fine-grained and difficult for humans, so we set the human scoring range to 1-5. Therefore, for comparability of GPT-4 scores and human scores in Table 2, we multiply human scores by 2 to match the range of GPT-4 scores.

## 3 Experiments

### 3.1 Experimental Settings

To better reflect the evaluation of FLUB's fallacy understanding ability of LLMs, we select some advanced LLMs that are widely used in the Chinese community: (1) **ERNIE-Bot** [15] is a series of closed-sourced commercial LLMs released by Baidu. We evaluate the three latest chat models, including ERNIE-Bot-3.5, ERNIE-Bot-3.5-Turbo, and ERNIE-Bot-4.0. (2) **ChatGPT** [16] ChatGPT is undoubtedly the hottest model developed by OpenAI. We evaluate GPT-3.5-Turbo and GPT-4-Turbo. (3) **ChatGLM3** [17] is the latest open-sourced model of the ChatGLM which is a series of bilingual LLMs. We evaluate the only open-sourced parameter size of ChatGLM3-6B. (4) **Qwen** [18] is the open-sourced LLMs developed by the Alibaba Group. We select three chat Qwen models, including Qwen-7B-Chat, Qwen-14B-Chat, and Qwen-72B-Chat. (5) **Yi** [19] series models are open-sourced LLMs trained from scratch by 01-AI. In our experiments, we select Yi-6B-Chat and Yi-34B-Chat to be evaluated on FLUB. (6) **Baichuan2** [20] has achieved the competitive performance of its size on many Chinese benchmarks. We select Baichuan2-7B-Chat and Baichuan2-13B-Chat.

When running LLMs inference, for closed-sourced LLMs, we access corresponding models via the official APIs. Meanwhile, open-sourced models are deployed on 1 to 4 NVIDIA A100 GPUs depending on their parameter size.

### 3.2 Automatic Evaluation Results

The main results are presented in Table 1 and we have the following insights:

1. **For the difficulty of different tasks**, the Answer Selection task is the simplest, which shows that LLMs should have a certain ability to distinguish right from wrong when seeing correct and wrong answers. However, we also see that the performance of all models on

the Cunning Type Classification task is unsatisfactory, with F-1 scores below 15.0, and some models even perform below random performance. This deficiency may stem from the models' limited capability to comprehend the semantics of various cunning types.

2. **For the connection between different tasks**, the comparative outcomes among different models across the three tasks are not consistent. Nevertheless, models that exhibit superior performance in the Answer Selection task tend to generate more plausible explanations. This phenomenon reminds us that there is a close relationship between the Answer Selection task and the Fallacy Explanation task. The interaction between these two tasks is very critical for improving the fallacy understanding ability of LLMs.

3. **For the model performance of different scale parameters**, overall, models of larger scale are better equipped to understand cunning texts, which aligns with intuitive expectations. Of course, there are exceptions. We find that for the Qwen and Yi models, as the parameter size increases, the performance of the Cunning Type Classification task decreases. This is because this task requires a deep understanding of the Chinese language, especially the popular Internet language, and we observe that as the Qwen and Yi models become larger, their ability to understand special Internet language becomes poorer. Besides, the another reason for the poor cunning type classification performance of the models is that they cannot accurately understand the defined types. Therefore, how to improve the perception ability of LLMs for the cunning types will be the key challenge to improving the performance of LLMs on the cunning classification task.

4. **For the impact of Chain-of-Thought**, to our surprise, Chain-of-Thought (CoT) does not bring stable improvements to LLMs' fallacy understanding ability. Especially for the Answer Selection and Fallacy Explanation tasks, CoT even has negative impacts on some models. We think there are two main reasons for this phenomenon: (1) We notice that when the model size exceeds 10B, CoT still has positive effects on these two tasks. This reflects the challenge of our tasks, which makes CoT unable to stimulate the small models to have sufficient capabilities to cope with them. (2) For traditional QA tasks (such as commonsense reasoning, mathematical reasoning, etc.), CoT can improve performance because these tasks themselves are relatively logical, and the process of solving their questions can be modeled as the logical reasoning process. Unlike these tasks, our tasks are not very logical problems but require more intuition about the language. Hence, adding intermediate steps by the CoT has no significant effect on our tasks. In summary, our proposed tasks deserve further research to improve the fallacy understanding ability of LLMs.

5. **For the overall performance**, considering that the performance values of the three sub-tasks are very different, we use the geometric mean to balance the impact of each sub-task and avoid the excessive impact of a single extreme value on the overall performance. We see that the overall performance of each model is basically consistent with common sense, that is, the larger the model, the better the performance, and CoT also brings positive effects. This shows that FLUB is of high quality and suitable to measure the fallacy understanding ability of LLMs from an overall perspective.

6. **For the human performance**, we see that humans perform well on the Answer Selection and Cunning Type Classification tasks, which reflects the considerable gap in fallacy understanding between human intelligence and LLMs. It also shows that our proposed new benchmark and tasks are conducive to further promoting the progress of LLMs. Note that the reason why the Fallacy Explanation task is not suitable for evaluating human performance is that its automatic evaluation indicator is the GPT-4 Score. We think that using GPT-4 to evaluate explanations written by humans is unreasonable and unnecessary.

## 3.3  The Impact of In-context Learning

We select 5 high-performing LLMs to study the impact of in-context learning on LLMs' fallacy understanding ability. Demonstrations used for in-context learning are randomly selected. As shown in Figure 3, unlike Chain-of-Thought which has no stable positive effect, the LLMs' performance with in-context learning is basically on the rise as demonstrations increase. This indicates that letting LLMs see more examples can improve their fallacy understanding ability, but the number of examples must be large enough because we have also seen that when only one shot example is added, the performance of LLMs sometimes declines compared to the zero-shot cases.

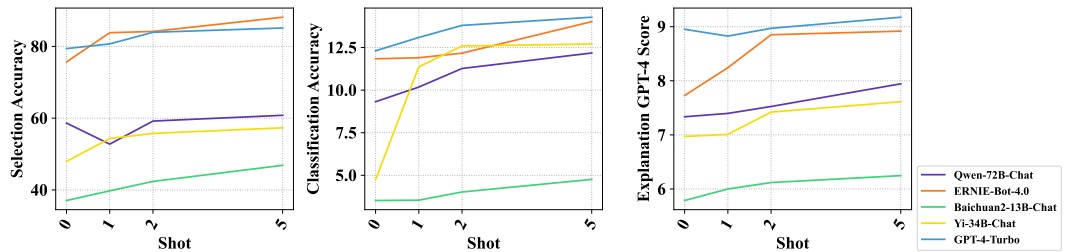

Figure 3: The results of in-context learning with 0/1/2/5-shots demonstrations.

Table 2: Human evaluation on the explanation task. Note that we multiply the human results by 2 to normalize their range to be the same as the GPT-4 results' range. The reported correlations are Spearman's rank correlation coefficients. All correlations are extremely significant with $p < 0.01$.

| Models | Human | GPT-4 | Correlation |
|---|---|---|---|
| GPT-4-Turbo | 7.12 | 8.60 | 0.57 |
| ERNIE-Bot-4.0 | 5.82 | 7.20 | 0.71 |
| Qwen-72B-Chat | 5.74 | 7.82 | 0.42 |
| Yi-34B-Chat | 5.42 | 6.44 | 0.74 |
| Baichuan2-13B-Chat | 4.42 | 5.84 | 0.63 |
| Overall | - | - | 0.69 |

## 3.4 Human Evaluation of Explanation

To verify the effectiveness of our designed automatic GPT-4 score for Task 3, we randomly select 50 data samples from FLUB, along with outputs from 5 high-performing LLMs for human evaluation by our contracted annotators. From the human evaluation results in Table 2, we observe that:

1. The overall correlation coefficient between the automatic and human evaluation is 0.69, indicating a high consistency between GPT-4 scores and human preferences. Besides, the correlation results also verify the effectiveness of our designed GPT-4 score for Task 3.

2. Both automatic and human evaluations exhibit a broadly consistent ranking across the selected five models. The GPT-4-Turbo achieves superior performance over all other models. In contrast, human annotators perceive marginal performance disparities among ERNIE-Bot-4.0, Qwen-72B-Chat, and Yi-34B-Chat models.

3. From the human evaluation results, except for GPT-4-Turbo, which can exceed the passing score of 6, the performance of other LLMs is still not ideal, which shows that the community still needs to further study how to improve the fallacy understanding ability of LLMs.

## 3.5 Case Study

To analyze FLUB's challenges, we conduct case studies on the two advanced models with better performance in the fallacy explanation task in Table 3. From the first case, we see that GPT-4-Turbo gives a relatively perfect explanation, while ERNIE-Bot-4.0's answer does not explain the causal relationship clearly although its final conclusion is correct. According to ERNIE-Bot-4.0's explanation, if the egg is added with water, it can be restored. This is obviously wrong. In the second case which is more difficult, both ERNIE-Bot-4.0 and GPT-4-Turbo easily fail when facing these cunning texts. Specifically, ERNIE-Bot-4.0 follows the trap of the input text, not clearly stating that "putting heads on the shore" is an impossible operation, but giving a dumbfounding explanation. In comparison, GPT-4-Turbo's performance is slightly better, but it does not perceive the trap in the input text at all, resulting in an answer that is not what is questioned. It can be seen from these two cases that LLMs' ability to handle cunning texts is still insufficient.

Table 3: We mark correct/partially correct/wrong explanations in green/orange/red.

| Input Text 1 | 有个问题，鸡蛋干加水会变成鸡蛋吗？
I have a question, will dried eggs turn into eggs if water is added to them? |
|---|---|
| Correct Answer | 鸡蛋干是将鸡蛋浓缩加工，而不是表面上"鸡蛋缺水变干"的意思，所以鸡蛋干加水不会变成鸡蛋。
Dried eggs are concentrated and processed eggs, rather than the superficial meaning of "eggs become dry due to lack of water", so adding water to dried eggs will not turn them into eggs. |
| ERNIE-Bot-4.0 | ...鸡蛋干是鸡蛋的加工制品，不是鸡蛋。加水不能使鸡蛋干还原成鸡蛋...
Dried eggs are processed products of eggs, not eggs. Adding water will not
turn the dried eggs back into eggs. |
| GPT-4-Turbo | ...鸡蛋干是经过脱水处理的鸡蛋产品，加水只能使其回软，但不能恢复到原始的鸡蛋状态...
Dried eggs are dehydrated egg products. Adding water can only soften them, but cannot return
them to their original egg state. |
| Input Text 2 | 游泳初学者怕呛死，下水前能先把头放岸边吗？
Beginner swimmers are afraid of choking. Can they put their heads on the shore before entering the water? |
| Correct Answer | 头是人身体的一部分，如果头离开了身体，不用呛死人就已经死亡了。
The head is part of the human body. If the head leaves the body, the person will die without choking him. |
| ERNIE-Bot-4.0 | ...将头部放在岸边并不是一种标准的游泳训练做法。游泳初学者下水前不应将头放在岸边...
Keeping your head on the bank is not a standard swim training practice. Beginner swimmers
should not put their heads on the shore before entering the water. |
| GPT-4-Turbo | ...游泳初学者在下水前可以先在岸边练习将头部放入水中并慢慢学会控制呼吸...
Beginner swimmers can practice putting their heads in the water on the shore and
slowly learn to control their breathing before entering the water. |

# 4 Related Work

**Reasoning Evaluation of LLMs**    Our FLUB is for evaluating the fallacy understanding ability of LLMs, which is closely related to the reasoning of LLMs [21–23]. Therefore, we first review related works on the commonsense and logical reasoning of LLMs. **Commonsense Reasoning**: Existing commonsense reasoning benchmarks include CommonsenseQA [24], PIQA [25], Social IQA [26], HellaSWAG [27], and MCTACO [28]. Their task is presented in the form of multiple-choice questions. The recent LLMs reasoning evaluation works [29, 30] have demonstrated that LLMs represented by ChatGPT often cannot accurately utilize commonsense knowledge for the reasoning process. **Logical Reasoning**: For logical reasoning data resources, they can be mainly divided into two categories: Natural Language Inference [31–33] and Multiple-Choice Reading Comprehension [34–37]. [38] show that logical reasoning is very challenging for LLMs, especially for out-of-distribution data samples. In summary, research on reasoning ability is the focus of the LLMs-centric research.

**Humor in NLP**    We notice that some samples in FLUB contain humorous expressions. Therefore, NLP research on humor [3, 39] is instructive for future exploration on FLUB. Particularly, as a representative humor task, the word game task with puns as the core has been continuously paid attention to by researchers [40–43]. According to our statistics, a large proportion of FLUB are cunning texts belonging to word games. Therefore, we believe that how to improve the humor recognition and processing capabilities of LLMs is also the key to improving the performance of LLMs on FLUB.

# 5 Limitations

One limitation of FLUB may be that it consists of Chinese data. In particular, many of the cunning texts in FLUB have certain Chinese cultural and language characteristics as backgrounds, which places extremely high demands on LLMs' knowledge storage. However, as a community that cannot be ignored in NLP, the development of Chinese LLMs has been devoted by generations of researchers. In addition, using GPT-4 to evaluate the output of other LLMs is already a widely used method. Although using GPT-4 to evaluate GPT-4 may be biased, using GPT-4 to evaluate other models still has reference value. In addition, not only for the tasks we proposed, but also for other tasks, the community is still actively exploring how to effectively evaluate LLMs. As a temporary compromise, we remind readers that they should interpret the GPT-4 scores carefully.

## 6 Ethics Statement

In this paper, we present a new benchmark, FLUB. We have described the details of the collection, preprocessing, and annotation of FLUB. And we ensure that no infringement or unethical behavior occurred during the dataset construction. In terms of the data itself, to ensure that the dataset we need to release in the future meets ethical requirements, we spend lots of energy on data anonymization, data desensitization, improper data cleaning, etc. Besides, the cunning texts we are concerned about come from daily life and are very common. Therefore, the new research direction and tasks we propose will not cause harm to human society.

## 7 Conclusion

In this work, we construct FLUB, a high-quality benchmark consisting of cunning texts designed to evaluate the fallacy understanding ability of LLMs. Furthermore, we evaluate advanced LLMs on FLUB. Detailed analyses indicate FLUB is very challenging and of great research value. To date, most existing LLMs still can not understand the fallacy well, which results in them being far from dealing with complex problems in the real world as easily as humans. We believe that the benchmark and the research direction we provide are valuable for the LLMs community.

## Acknowledgments and Disclosure of Funding

This research is supported by the National Natural Science Foundation of China (Grant No. 62276154), the Natural Science Foundation of Guangdong Province (Grant No. 2023A1515012914 and 440300241033100801770), Shenzhen Science and Technology Program (Grant No. WDZC20231128091437002), Basic Research Fund of Shenzhen City (Grant No. JCYJ20210324120012033 and GJHZ202402183000101), the Major Key Project of PCL for Experiments and Applications (PCL2021A06), the Guangdong Provincial Department of Education Project (Grant No.2024KQNCX028); Scientific Research Projects for the Higher-educational Institutions (Grant No.2024312096), Education Bureau of Guangzhou Municipality; Guangzhou-HKUST(GZ) Joint Funding Program (Grant No.SL2024A03J01201), Education Bureau of Guangzhou Municipality; China Association for Science and Technology (Grant No.XMSB20240711064). This work is also supported in part by NSF under grants III-2106758, and POSE-2346158.

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

Figure 4: Our designed prompts without the Chain-of-Thought idea. Task 3(a) is for the texts that are not expressed in the form of inquiries. Task 3(b) is for inquiries.

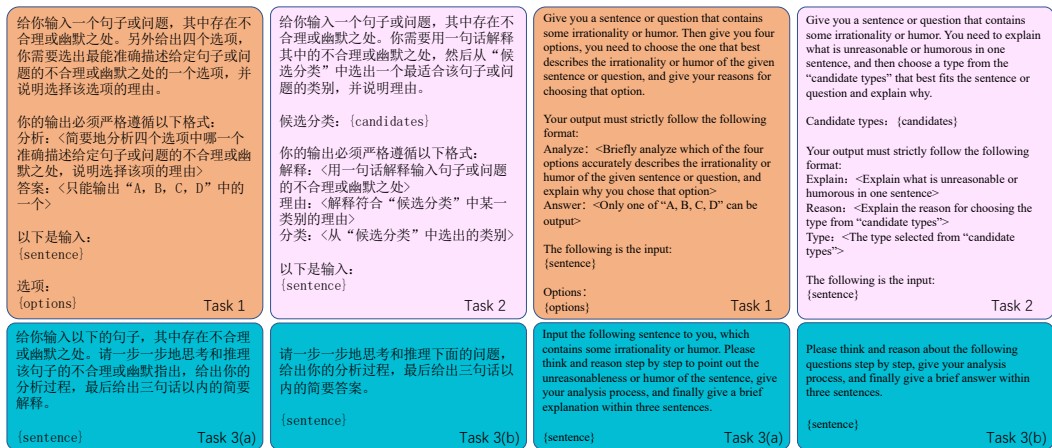

Figure 5: Our designed prompts with the Chain-of-Thought idea. Task 3(a) is for the texts that are not expressed in the form of inquiries. Task 3(b) is for inquiries.

## A  Our Designed Prompts for FLUB

Our designed prompts without Chain-of-Thought for FLUB are shown in Figure 4. And the Chain-of-Thought prompts for Task 1 and Task 2 are presented in Figure 5. For the convenience of reading, we provide both prompts and their English translations. Note that we use Chinese prompts in specific experiments because our data samples are in Chinese.

## B  Automatic Evaluation Prompt in Task 3

To enable GPT-4 can automatically score explanations generated by other LLMs according to the settings of Task 3, we carefully design the corresponding prompt, as shown in Figure 6.

## C  Scoring Guidelines in Task 3

**Scoring Objective**    For the LLMs' output response to each input cunning text, please refer to the given annotated answer to score the output response. This score will be used to measure the quality of the model response and the degree of matching with the answer.

**Scoring Rules**    The scoring values are defined as {1, 2, 3, 4, 5}. The higher the score, the higher the quality of the model output. **The specific definitions and examples of each score are as follows:**

```
[Instruction]
Please act as an impartial judge and evaluate the quality of the response provided by an
AI assistant to the user question displayed below. Your evaluation should consider
correctness and helpfulness. You will be given a reference answer and the assistant's
answer. Begin your evaluation by comparing the assistant's answer with the reference
answer. Identify and correct any mistakes. Be as objective as possible. After providing
your explanation, you must rate the response on a scale of 1 to 10 by strictly following
this format: \"[[rating]]\", for example: \"Rating: [[5]]\".

[Question]
{question}

[The Start of Reference Answer]
{answer}
[The End of Reference Answer]

[The Start of Assistant's Answer]
{response}
[The End of Assistant's Answer]
```

Figure 6: The designed prompt of automatic evaluation for Task 3.

## C.1    Definition and Examples of Score 1

**Definition**    The LLM's response doesn't point out the irrationality or humor in the given text, which is completely unrelated to the correct answer.

### C.1.1    The First Example of Score 1

**Input Text:**    Since instant noodles can be eaten dry, then dry crispy noodles can also be boiled? (既然方便面可以干吃，那干脆面也可以水煮？)

**Model Response:**    Yes, dry crispy noodles can also be boiled. Boil in boiling water until softened and serve. (是的，干脆面也可以水煮。将其放入沸水中煮至软化即可食用。)

**Correct Answer:**    Dry crispy noodles are crispy and delicious ready-to-eat snack food. You can eat them as soon as you open the package. What people need is its crispness. So its taste will be lost after being boiled. Therefore, crispy noodles cannot be boiled. (干脆面是一种香脆可口的即食型休闲食品，打开包装就可以吃，吃的就是它的香脆，水煮之后会失去它的口感。所以干脆面不可以水煮。)

### C.1.2    The Second Example of Score 1

**Input Text:**    To say that Ruozhi Bar is all a bunch of retards is that considered a compliment or an insult? (说弱智吧都是一帮弱智，算夸奖还是辱骂？)

**Model Response:**    This sentence is insulting because it uses the derogatory term "retarded" to describe a group of people and expresses contempt and insult to them. (这句话是辱骂，因为它使用贬义词汇"弱智"来形容一群人，表达了对他们的轻视和侮辱。)

**Correct Answer:**    "Ruozhi Bar" is the name of the post bar, and the IQ of the members inside has nothing to do with what the post bar name is. "Retarded" means intelligence below the normal level, and to say 'Ruozhi Bar" is a bunch of retarded is an insult. ("弱智吧"是贴吧名，里面成员的智商与贴吧名是什么无关。"弱智"的意思是智力低于正常水平，说弱智吧都是一帮弱智是一种辱骂。)

## C.2    Definition and Examples of Score 2

**Definition**    Although LLM's response points out the unreasonable/humorous points in the given text, the types and key points pointed out are inconsistent with the opinions of explanations.

### C.2.1 The First Example of Score 2

**Input Text:** Dreams are all reversed. I dream of myself looking in the mirror, so I am the one in the mirror. (梦都是反的，我梦到自己照镜子，所以我才是镜子里的人。)

**Model Response:** The humor of this sentence lies in the literal interpretation of the saying "Dreams are all reversed". Usually, people think that the scene in a dream is the opposite of reality, but in this sentence, the author dreams of himself looking in the mirror, if we understand that "Dreams are all reversed", then the author is actually a reflection in the mirror, which is obviously an absurd interpretation of reality, and therefore has a sense of humor. (这个句子的幽默之处在于对"梦都是反的"这一说法的字面解释。通常，人们认为梦中的情景与现实相反，但在这个句子中，作者梦到自己照镜子，如果按照"梦都是反的"来理解，那么作者实际上是镜子里的倒影，这显然是对现实的荒诞解释，因此具有幽默感。)

**Correct Answer:** "Dreams are all reversed" means that the dream is opposite to reality. If "I dream of myself looking in the mirror" is contrary to reality, the reality should be "I do not look in the mirror in reality", rather than "I dream of myself looking in the mirror, and I am the person in the mirror". ("梦都是反的"是指梦境与现实是相反的，"我梦到自己照镜子"与现实相反，应该是"现实中没有照镜子"，而不是"我梦到自己照镜子，反过来我是镜中人"。)

### C.2.2 The Second Example of Score 2

**Input Text:** If all three of Nezha's heads suffered from cerebral thrombosis for ten years, did he have cerebral thrombosis for ten years or thirty years? (哪吒三头都得了十年脑血栓那是算十年脑血栓还是三十年脑血栓？)

**Model Response:** Nezha is a fictional character in Chinese mythology that does not exist. Cerebral thrombosis is a cerebrovascular disease, usually caused by blood clotting in the blood vessels of the brain resulting in brain hypoxia and ischemia. Therefore, the claim that Nezha's three heads have cerebral thrombosis is not realistic, let alone calculate whether it is ten years or thirty years. (哪吒是中国神话中的虚构人物，没有真实存在。脑血栓是一种脑血管疾病，通常是由于脑部血管内的血液凝固形成的血栓导致脑部缺氧缺血。因此，哪吒三头得脑血栓的说法是不符合实际的，更不用计算是十年还是三十年。)

**Correct Answer:** Nezha is an individual, assuming Nezha's three heads all have cerebral thrombosis, it should be something that happens at the same time, and will not produce accumulation, so it should not be 30 years. (哪吒是一个个体，假设哪吒的三个头都得了脑血栓，那应该是同时发生的事情，并不会产生累加，所以不应该是三十年。)

### C.3 Definition and Examples of Score 3

**Definition** The LLM's response correctly points out the type of irrationality/humor, but the key points of irrationality/humor pointed out by the correct answer are inconsistent. Alternatively, the response is consistent with the key points of irrationality/humor pointed out by the correct answer, but the judgment on the type of cunning text is inconsistent.

### C.3.1 The First Example of Score 3

**Input Text:** The drug is partially poisonous with side effects, so we should not take the drug in parts. (是药三分毒，所以药不能分三次喝。)

**Model Response:** The humor of this sentence lies in the misunderstanding of the literal and actual meaning of the idiom "The drug is partially poisonous with side effects". This idiom is meant to say that any drug has potential side effects, while the sentence misunderstands "partially" to mean that the drug should not be taken in parts, and thus draws the absurd conclusion that the drug cannot be taken in three separate doses, which is obviously a comical distortion of the original meaning. (这句子的幽默之处在于对成语"是药三分毒"的字面意思和实际含义的误解。成语本意是说任何药物都有潜在的副作用，而句子却把"三分"误解为药物应该分成三个部分服用，因此得出了一个荒谬的结论，即药物不能按照三次分开服用，这显然是对原意的一种滑稽曲解。)

**Correct Answer:** "The drug is partially poisonous with side effects" refers to any drug will have certain side effects if used improperly. "Partially" indicates that to a certain extent, there is no meaning of "in parts". Therefore, the conclusion "we should not take the drug in parts" cannot be drawn from this sentence. ("是药三分毒"指的是任何的药品如果使用不当的话都会存在一定的副作用，"三分"表示的是在一定程度上，并没有分三次的意思，所以"是药三分毒"这句话并不能得出"药不能分三次喝"这一结论。)

### C.3.2 The Second Example of Score 3

**Input Text:** Did a knee-jerk reflex test on a friend, and the result was that his leg never lifted again after the hammer was knocked down. This reflex arc is too long! (给朋友做膝跳测试，结果锤下去之后他的腿再也没抬起过，这反射弧也太长了吧！)

**Model Response:** The knee-jerk reflex test is a simple test to check the function of the nervous system. Normally, a light tap on the tendon below the knee will cause the leg to make a quick leg lift motion. This sentence humorously exaggerates the amount of time the friend does not respond, indicating that the reflex arc is unusually long, when in fact, if there is really no response, it may indicate a neurological problem rather than the reflex arc length. In addition, using "never lifted again" has a more humorous effect because it indicates a permanent state, which is impossible in reality. (膝跳反射测试是一种检查神经系统功能的简单测试，正常情况，轻轻敲击膝盖下方的腱会导致腿部做出快速的抬腿动作。这句话幽默地夸张了朋友没有反应的时间，暗示反射弧异常地长，实际上如果真的没有反应，那可能表明存在神经系统问题，而不是反射弧长。此外，用"再也没抬起过"增加了幽默效果，因为它暗示了一种永久性的状态，这在现实中是不可能的。)

**Correct Answer:** "His leg never lifted again after the hammer" indicates that the friend's leg was broken, and is not related to the length of the reflex arc, here is actually a humorous way to express the matter of "breaking the friend's leg". ("锤下去之后他的腿再也没抬起过"说明把朋友的腿打断了，与反射弧长短没有关系，这里实际上是用幽默的方式将"把朋友腿打断"表达出来。)

## C.4 Definition and Examples of Score 4

**Definition** The response correctly points out the type of irrationality/humor in the input text, and the key points pointed out are correct, but the interpretation is not consistent with the correct answer.

### C.4.1 The First Example of Score 4

**Input Text:** In a vegetable market, one finds vegetative persons instead. So it must have been a job market before. (去素材库发现几个植物人躺在里面，原来之前这里是人材市场。)

**Model Response:** The humor of this sentence lies in the pun on the word "vegetative". Usually, "vegetative" refers to patients who have fallen into a persistent vegetative state due to a major illness or injury, but here it is used to refer to "vegetable" in the literal sense, i.e., real plants. At the same time, "job market" usually refers to a market for hiring "people" with the appropriate skill, but here it is used teasingly to refer to a market with vegetative "people". This linguistic coincidence creates an absurd humorous effect. (这个句子的幽默之处在于使用了"植物人"这个词语的双关意味。通常，"植物人"指的是那些因为重大疾病或伤害而陷入持续植物状态的病人，但在这里它被用来指代字面意义上的"植物"，即真正的植物。同时，"人材市场"通常是指招聘人才的市场，而在这里它被戏谑地用来指一个植物市场，这种语言上的巧合创造了一种荒谬的幽默效果。)

**Correct Answer:** The vegetable market refers to the market where vegetables are sold. "Vegetative" is a special human state similar to the living state of plants, and the human body cannot lie in the vegetable market. (素材库是指销售蔬菜的市场。"植物人"是与植物生存状态相似的特殊的人体状态，人体是无法躺进素材库中的。)

### C.4.2 The Second Example of Score 4

**Input Text:** If you drink 82 Lafite, then you'll live at least 82 years. (喝了82年拉菲至少能活82岁。)

**Model Response:** The humor or irrationality of the sentence is that it establishes a false cause-and-effect relationship, implying that drinking a certain vintage of wine guarantees living to the same age as the vintage, which obviously has no scientific basis. In fact, a person's lifespan is affected by many factors, including genetics, lifestyle, medical conditions, and so on, rather than a single diet. In addition, this statement also carries a kind of humorous exaggeration, because it uses an absurd logic to attract attention and increase the interest of the words. (该句子的幽默或不合理之处在于它建立了一种错误的因果关系，暗示喝了某年份的葡萄酒能保证活到与年份相同的年龄，这显然没有科学依据。实际上，人的寿命受多种因素影响，包括遗传、生活方式、医疗条件等，而非单一的饮食习惯。此外，这种说法也带有一种幽默的夸张，因为它用一种荒谬的逻辑来吸引注意，增加话语的趣味性。)

**Correct Answer:** "82 Lafite" is a network buzzword that originated from scenes of opening Lafite wine in movies and TV series. Since the Lafite red wine produced in 1982 is of good quality and expensive, "82 Lafite" is synonymous with high standards to set off the character's extravagant spending and taste, rather than "Lafite has been drunk for 82 years". (82年的拉菲，网络流行语，源自影视剧中开拉菲的桥段，由于82年的拉菲红酒品质好、价格昂贵，所以"82年的拉菲"是高规格的代名词，来衬托人物的挥金如土和品味，而不是"喝了八十二年的拉菲"的意思。)

## C.5 Definition and Examples of Score 5

**Definition** The LLM's response and the correct explanation are basically matched, the response correctly points out the type of unreasonable/humorous points in the input text, the key points are correct, and the explanation is basically consistent with the explanation.

### C.5.1 The First Example of Score 5

**Input Text:** If a police officer commits a crime and turns himself in, is it considered to have made meritorious contributions? (警察犯了事去自首算不算戴罪立功啊？)

**Model Response:** "Meritorious service" usually refers to the behavior of criminals who actively cooperate with the police to detect other cases or fight criminal activities after committing a crime, so as to reduce their own punishment. If the police officer has committed a crime and turned himself in, and there is no further meritorious service, then this can usually only be counted as surrender, and cannot be directly referred to as meritorious service. However, surrendering may lead to reduced punishment according to relevant legal provisions. ("戴罪立功"通常指的是犯罪分子在犯罪后，积极配合警方侦破其他案件或对抗犯罪活动，从而争取减轻自己刑罚的行为。如果警察犯了罪后去自首，并且没有进一步的立功表现，那么这通常只能算作自首，并不能直接称为戴罪立功。不过，自首可能会根据相关法律规定减轻处罚。)

**Correct Answer:** According to the provisions of Article 67 of the Criminal Law, "Surrender" refers to the act of automatically surrendering to the police after committing a crime and truthfully confessing one's crime. If a criminal suspect or defendant under compulsory measures or a criminal serving a sentence truthfully confesses other crimes that are not yet known to the judicial organs, he shall be regarded as surrendering. "Meritorious service" means that the meritorious service performance in criminal law refers to the performance of criminal suspects providing important clues, exposing others' crimes, etc., and can be mitigated after verification. Surrender confession by oneself is one's own crime, while meritorious service is to expose the crime of others, so the police committing the crime to surrender is not regarded as meritorious service. (根据刑法第67条的规定，"自首"是指犯罪以后自动投案，如实供述自己的罪行的行为。被采取强制措施的犯罪嫌疑人、被告人和正在服刑的罪犯，如实供述司法机关还未掌握的本人其他罪行的，以自首论。"戴罪立功"就是指刑法中的立功表现是指犯罪嫌疑人提供重要线索、揭发他人犯罪行等的表现，经查实的可以获得减轻处罚。自首供述的是本人的罪行，戴罪立功是揭发他人罪行，所以条子犯了事去自首不算戴罪立功。)

### C.5.2 The Second Example of Score 5

**Input Text:** Officer Wang shot the hostage and saved the robber who could have been killed at any time. (王警官击毙人质，解救了随时可能被狙杀的劫匪。)

**Model Response:** The unreasonableness of this sentence lies in the fact that Officer Wang's actions are contrary to the duties of ordinary police officers. Usually, the task of the police is to protect the hostages and try to arrest the criminals, not to kill the hostages to protect the criminals. Thus, the sentence reverses the duties of the police with an absurd logic that produces a humorous effect. (这句子的不合理之处在于王警官的行动与一般警察的职责相反。通常情况下，警察的任务是保护人质并尽量逮捕犯罪分子，而不是击毙人质来保护犯罪分子。因此，这个句子以一种荒谬的逻辑颠倒了警察的职责，产生了幽默效果。)

**Correct Answer:** As a police officer, what Wang should do is shoot the robber to save the hostage, not shoot the hostage to save the robber. The fact that Wang killed the hostage by mistake is described humorously. (王警官作为警察，应该做的是击毙劫匪解救人质，而不是击毙人质解救劫匪，这里用幽默的方式阐述了王警官误杀人质这一事实。)

# D   Datasheet For FLUB

## D.1   Motivation

1. **For what purpose was the dataset created?** Was there a specific task in mind? Was there a specific gap that needed to be filled? Please provide a description.

   - In this work, we aim to challenge the reasoning and understanding abilities of LLMs by proposing the FLUB containing cunning texts that are easy for humans to understand but difficult for models to grasp. Specifically, we design three tasks with increasing difficulty to test whether the LLMs can understand the fallacy and solve the "cunning" texts: Answer Selection, (2) Cunning Type Classification, (3) Fallacy Explanation. We hope and believe that our proposed FLUB and all our findings are crucial for LLMs to comprehend the fallacy and handle cunning texts in the real world.

2. **Who created the dataset (e.g., which team, research group) and on behalf of which entity (e.g., company, institution, organization)?**

   - The dataset is presented by Tsinghua Knowledge Engineering Laboratory (SZ).

3. **Who funded the creation of the dataset?** If there is an associated grant, please provide the name of the grantor and the grant name and number.

   - This work is sponsored by NSFC, Guangdong Province, Shenzhen City, Peng Cheng Laboratory, and Tsinghua Univerisity.

4. **Any other comments?**

   - No.

## D.2   Composition

5. **What do the instances that comprise the dataset represent (e.g., documents, photos, people, countries)?** Are there multiple types of instances (e.g., movies, users, and ratings; people and interactions between them; nodes and edges)? Please provide a description.

   - All the instances in FLUB are represented by texts. We make our benchmark openly available on the GitHub page (https://github.com/THUKElab/FLUB).

6. **How many instances are there in total (of each type, if appropriate)?**

   - FLUB includes 834 instances. For fine-grained cunning types, "False Analogy" has 11 instances, "Lame Jokes" has 44 instances, "Phonetic Error" has 5 instances, "Ambiguity" has 35 instances, "Paradox" has 29 instances, "Factual Error" has 12 instances, "Reasoning Error" has 445 instances, "Word Game" has 239 instances, and "Undefined" has 14 instances.

7. **Does the dataset contain all possible instances or is it a sample (not necessarily random) of instances from a larger set?** If the dataset is a sample, then what is the larger set? Is the sample representative of the larger set (e.g., geographic coverage)? If so, please describe how this representativeness was validated/verified. If it is not representative of the larger set, please describe why not (e.g., to cover a more diverse range of instances, because instances were withheld or unavailable).

- FLUB has contained all possible instances, because we have tried our best to collect as much data as possible from "Ruozhiba" and conducted strict manual annotation.

8. **What data does each instance consist of? "Raw" data (e.g., unprocessed text or images) or features?** In either case, please provide a description.

   - Each instance consists of the input cunning text, cunning type, fallacy explanation, candidate answers, and the corresponding correct option, as illustrated in Figure 1b.

9. **Is there a label or target associated with each instance?** If so, please provide a description.

   - There is a cunning type for each instance, which describes the cunning type of each input text.

10. **Is any information missing from individual instances?** If so, please provide a description, explaining why this information is missing (e.g., because it was unavailable). This does not include intentionally removed information, but might include, e.g., redacted text.

    - No.

11. **Are relationships between individual instances made explicit (e.g., users' movie ratings, social network links)?** If so, please describe how these relationships are made explicit.

    - Not applicable.

12. **Are there recommended data splits (e.g., training, development/validation, testing)?** If so, please provide a description of these splits, explaining the rationale behind them.

    - No, because FLUB is a benchmark test set, all its instances are used for testing LLMs, regardless of training/validation.

13. **Are there any errors, sources of noise, or redundancies in the dataset?** If so, please provide a description.

    - No.

14. **Is the dataset self-contained, or does it link to or otherwise rely on external resources (e.g., websites, tweets, other datasets)?** If it links to or relies on external resources, a) are there guarantees that they will exist, and remain constant, over time; b) are there official archival versions of the complete dataset (i.e., including the external resources as they existed at the time the dataset was created); c) are there any restrictions (e.g., licenses, fees) associated with any of the external resources that might apply to a dataset consumer? Please provide descriptions of all external resources and any restrictions associated with them, as well as links or other access points, as appropriate.

    - FLUB is self-contained.

15. **Does the dataset contain data that might be considered confidential (e.g., data that is protected by legal privilege or by doctor–patient confidentiality, data that includes the content of individuals' non-public communications)?** If so, please provide a description.

    - No.

16. **Does the dataset contain data that, if viewed directly, might be offensive, insulting, threatening, or might otherwise cause anxiety?** If so, please describe why.

    - No. We have conducted a strict data cleaning process to ensure that FLUB does not contain unethical data.

17. **Does the dataset identify any subpopulations (e.g., by age, gender)?** If so, please describe how these subpopulations are identified and provide a description of their respective distributions within the dataset.

    - No.

18. **Is it possible to identify individuals (i.e., one or more natural persons), either directly or indirectly (i.e., in combination with other data) from the dataset?** If so, please describe how.

    - No.

19. **Does the dataset contain data that might be considered sensitive in any way (e.g., data that reveals race or ethnic origins, sexual orientations, religious beliefs, political opinions or union memberships, or locations; financial or health data; biometric or**

**genetic data; forms of government identification, such as social security numbers; criminal history)?** If so, please provide a description.

- No.

20. **Any other comments?**

- No.

### D.3 Collection Process

21. **How was the data associated with each instance acquired? Was the data directly observable (e.g., raw text, movie ratings), reported by subjects (e.g., survey responses), or indirectly inferred/derived from other data (e.g., part-of-speech tags, model-based guesses for age or language)?** If the data was reported by subjects or indirectly inferred/derived from other data, was the data validated/verified? If so, please describe how.

- We collect raw text data from "Ruozhiba" in Baidu Tieba, as described in Section 2.1. The data is directly observable at `https://github.com/THUKElab/FLUB`.

22. **What mechanisms or procedures were used to collect the data (e.g., hardware apparatuses or sensors, manual human curation, software programs, software APIs)? How were these mechanisms or procedures validated?**

- We use web crawlers to automatically crawl the raw data, and we perform manual filtering and filtering to validate the crawled data, as described in Section 2.1.

23. **If the dataset is a sample from a larger set, what was the sampling strategy (e.g., deterministic, probabilistic with specific sampling probabilities)?**

- Not applicable.

24. **Who was involved in the data collection process (e.g., students, crowdworkers, contractors) and how were they compensated (e.g., how much were crowdworkers paid)?**

- We hired crowdworkers to clean the raw data and paid each person $0.50 per piece of raw data.

25. **Over what timeframe was the data collected?** Does this timeframe match the creation timeframe of the data associated with the instances (e.g., recent crawl of old news articles)? If not, please describe the timeframe in which the data associated with the instances was created.

- The raw data of FLUB was collected in in October 2023. The task characteristics of FLUB are not time-sensitive, so the collection time is not associated with the data instances.

26. **Were any ethical review processes conducted (e.g., by an institutional review board)?** If so, please provide a description of these review processes, including the outcomes, as well as a link or other access point to any supporting documentation.

- Not applicable. Our data collection process does not involve human or animal experiments. In addition, according to the Baidu Bar agreement, the data on Baidu Tieba can be used for academic research free of charge and without liability. Therefore, our data collection process does not require the involvement of an ethical review board.

27. **Did you collect the data from the individuals in question directly, or obtain it via third parties or other sources (e.g., websites)?**

- We collect raw text data from "Ruozhiba" in Baidu Tieba, as described in Section 2.1.

28. **Were the individuals in question notified about the data collection?** If so, please describe (or show with screenshots or other information) how notice was provided, and provide a link or other access point to, or otherwise reproduce, the exact language of the notification itself.

- Not applicable. According to the Baidu Bar agreement, the data on Baidu Tieba can be used for academic research free of charge and without liability.

29. **Did the individuals in question consent to the collection and use of their data?** If so, please describe (or show with screenshots or other information) how consent was requested and provided, and provide a link or other access point to, or otherwise reproduce, the exact language to which the individuals consented.

- Yes. According to the Baidu Bar agreement, the data on Baidu Tieba can be used for academic research free of charge and without liability.

30. **If consent was obtained, were the consenting individuals provided with a mechanism to revoke their consent in the future or for certain uses?** If so, please provide a description, as well as a link or other access point to the mechanism (if appropriate).

    - Not applicable.

31. **Has an analysis of the potential impact of the dataset and its use on data subjects (e.g., a data protection impact analysis) been conducted?** If so, please provide a description of this analysis, including the outcomes, as well as a link or other access point to any supporting documentation.

    - Yes. The cunning texts we are concerned about come from daily life and are very common. Therefore, the new research direction and tasks we propose will not cause harm to human society.

32. **Any other comments?**

    - No.

## D.4  Preprocessing/cleaning/labeling

33. **Was any preprocessing/cleaning/labeling of the data done (e.g., discretization or bucketing, tokenization, part-of-speech tagging, SIFT feature extraction, removal of instances, processing of missing values)?** If so, please provide a description. If not, you may skip the remaining questions in this section.

    - Yes. We employ annotators to manually filter out irrelevant posts that do not present cunning texts. Since the collected original posts contain irrelevant content such as links and images, we also require annotators to extract the fallacious and illogical contents from the raw post and rewrite them into a complete sentence. Besides, it is worth noting that we carefully ensure that the texts in FLUB are ethical texts. This process includes user information anonymization, sensitive information removal, and filtering of impolite posts.

34. **Was the "raw" data saved in addition to the preprocessed/cleaned/labeled data (e.g., to support unanticipated future uses)?** If so, please provide a link or other access point to the "raw" data.

    - No.

35. **Is the software that was used to preprocess/clean/label the data available?** If so, please provide a link or other access point.

    - No.

36. **Any other comments?**

    - No.

## D.5  Uses

37. **Has the dataset been used for any tasks already?** If so, please provide a description.

    - No.

38. **Is there a repository that links to any or all papers or systems that use the dataset?** If so, please provide a link or other access point.

    - No.

39. **What (other) tasks could the dataset be used for?**

    - Based on our constructed FLUB and its annotation information, we design three tasks with increasing difficulty to test whether the LLMs can understand the fallacy and solve the "cunning" texts. Specifically, (1) Answer Selection: The model is asked to select the correct one from the four answers provided by FLUB for each input text. (2) Cunning Type Classification: Given a cunning text as input, the model is expected to directly identify its fallacy type defined in our scheme. (3) Fallacy Explanation: We

hope the model sees a cunning text and intelligently generates a correct explanation for the fallacy contained in the text, just like humans, without falling into its trap.

40. **Is there anything about the composition of the dataset or the way it was collected and preprocessed/cleaned/labeled that might impact future uses?** For example, is there anything that a dataset consumer might need to know to avoid uses that could result in unfair treatment of individuals or groups (e.g., stereotyping, quality of service issues) or other risks or harms (e.g., legal risks, financial harms)? If so, please provide a description. Is there anything a dataset consumer could do to mitigate these risks or harms?

   - No.

41. **Are there tasks for which the dataset should not be used?** If so, please provide a description.

   - According to the characteristics of the data in FLUB, it is known that in addition to the three benchmark tasks we designed, we think that it may also be suitable for improving the reasoning ability and humor ability of LLMs. Beyond that, FLUB may not be suitable for other tasks.

42. **Any other comments?**

   - No.

## D.6 Distribution

43. **Will the dataset be distributed to third parties outside of the entity (e.g., company, institution, organization) on behalf of which the dataset was created?** If so, please provide a description.

   - Yes, the dataset has been open-source.

44. **How will the dataset will be distributed (e.g., tarball on website, API, GitHub)?** Does the dataset have a digital object identifier (DOI)?

   - The data is available through `https://github.com/THUKElab/FLUB`.

45. **When will the dataset be distributed?**

   - The dataset has been open-source.

46. **Will the dataset be distributed under a copyright or other intellectual property (IP) license, and/or under applicable terms of use (ToU)?** If so, please describe this license and/or ToU, and provide a link or other access point to, or otherwise reproduce, any relevant licensing terms or ToU, as well as any fees associated with these restrictions.

   - FLUB is published under Creative Commons Attribution-NonCommercial 4.0 International (CC BY-NC 4.0), which means everyone can use this dataset for non-commercial research purposes.

47. **Have any third parties imposed IP-based or other restrictions on the data associated with the instances?** If so, please describe these restrictions, and provide a link or other access point to, or otherwise reproduce, any relevant licensing terms, as well as any fees associated with these restrictions.

   - We collect raw text data from "Ruozhiba" in Baidu Tieba. According to the Baidu Bar agreement, the data on Baidu Tieba can be used for academic research free of charge and without liability.

48. **Do any export controls or other regulatory restrictions apply to the dataset or to individual instances?** If so, please describe these restrictions, and provide a link or other access point to, or otherwise reproduce, any supporting documentation.

   - No.

49. **Any other comments?**

   - No.

### D.7 Maintenance

50. **Who will be supporting/hosting/maintaining the dataset?**

    - Tsinghua Knowledge Engineering Laboratory (SZ) will support hosting of the dataset.

51. **How can the owner/curator/manager of the dataset be contacted (e.g., email address)?**

    - The manager of FLUB can be contacted through:
    - Email (liyinghu20@mails.tsinghua.edu.cn)
    - GitHub issues (https://github.com/THUKElab/FLUB/issues).

52. **Is there an erratum?** If so, please provide a link or other access point.

    - There is no erratum for our first release. Errata will be documented on the dataset website as a future release.

53. **Will the dataset be updated (e.g., to correct labeling errors, add new instances, delete instances)?** If so, please describe how often, by whom, and how updates will be communicated to dataset consumers (e.g., mailing list, GitHub)?

    - Yes. Once any other researchers find that FLUB needs to be updated, we will immediately update it through GitHub.

54. **If the dataset relates to people, are there applicable limits on the retention of the data associated with the instances (e.g., were the individuals in question told that their data would be retained for a fixed period of time and then deleted)?** If so, please describe these limits and explain how they will be enforced.

    - Not applicable.

55. **Will older versions of the dataset continue to be supported/hosted/maintained?** If so, please describe how. If not, please describe how its obsolescence will be communicated to dataset consumers.

    - Yes. We will continue to support FLUB. Once any other researchers find that FLUB needs to be updated, we will immediately update it through GitHub.

56. **If others want to extend/augment/build on/contribute to the dataset, is there a mechanism for them to do so? If so, please provide a description. Will these contributions be validated/verified?** If so, please describe how. If not, why not? Is there a process for communicating/distributing these contributions to dataset consumers? If so, please provide a description.

    - Yes. Once any other researchers find that FLUB needs to be updated, after they contact us via email or GitHub, we will review the data they want to expand. After the new data passes review, we will immediately update it to GitHub.

57. **Any other comments?**

    - No.

## E    Metadata and Data Format of FLUB

### E.1    Croissant Metadata

To provide the key descriptive information of FLUB more clearly and improve the traceability and reproducibility of our data, we also provide the Croissant metadata of FLUB, please refer to the link https://github.com/THUKElab/FLUB/blob/main/FLUB_croissant_metadata.json for details.

### E.2    Data Format

Listing 1: The data format of our FLUB.

```
{
  "text": "The input cunning text",
  "is_question": "Is the input cunning text a question?",
```

```
 "type": "The cunning type of the input text for the Cunning Type    4
     Classification task.",
 "explanation": "The correct explanation of the input text for       5
    the Fallacy Explanation task.",
 "id": "The id of each data sample",                                  6
 "options": {                                                         7
   "A": "The candidate answer 1 for the input text (question)",       8
   "B": "The candidate answer 2 for the input text (question)",       9
   "C": "The candidate answer 3 for the input text (question)",      10
   "D": "The candidate answer 4 for the input text (question)"       11
 },                                                                  12
 "answer": "The correct answer for the Answer Selection (Multiple    13
     Choice) task."
}                                                                    14
```

## F Author Statement of FLUB

We, as the authors of the FLUB dataset, hereby declare the following:

1. **Responsibility Statement**: The creation, organization, and publication of FLUB are entirely our responsibility. We confirm that all data were legally obtained and do not infringe on the intellectual property or other rights of any third party. In the event of any disputes or legal liabilities arising from the use of this dataset, we, as the authors, will assume full responsibility.

2. **Data License**: This dataset is released under the following license: Creative Commons Attribution-NonCommercial 4.0 International (CC BY-NC 4.0). Users must comply with the terms of this license agreement when using this dataset. For detailed license terms, please refer to CC BY-NC 4.0.

3. **Data Integrity and Quality**: We have made every effort to ensure the integrity and quality of FLUB. However, due to the dataset's size and complexity, we cannot guarantee it to be completely error-free. If any errors or omissions are discovered, please contact us for corrections and updates.

4. **Ethical Statement**: We have strictly adhered to relevant ethical guidelines during the data collection and processing stages to ensure that the use of this dataset does not negatively impact or harm any individual or organization.

