# OpenReview forum: "When LLMs Meet Cunning Texts: A Fallacy Understanding Benchmark for Large Language Models"
_NeurIPS.cc/2024/Datasets_and_Benchmarks_Track — NeurIPS 2024 Track Datasets and Benchmarks Poster_

### Official Review · Reviewer_SvrS · 2024-07-07
**When LLMs Meet Cunning Texts: A Fallacy Understanding Benchmark for Large Language Models**

**Rating:** 7
**Confidence:** 4
**Correctness:** Yes.
**Clarity:** Yes.

**Review:**

Please see above.

**Strengths:**

1. FLUB contains cunning texts that are designed to be challenging for LLMs by incorporating elements such as misleading information, wrong premises, intentional ambiguity, and other fallacies.
2. There are some good insights from the empirical results, such as the poor performance of LLMs in perceiving fallacy types, the non-linear relationship between the size of LLMs and their performance, and the significance of techniques like Chain-of-Thought and In-context Learning.

**Additional Feedback:**

NA.

**Documentation:**

Yes.

**Ethics:**

No.

**Limitations:**

Yes.

**Opportunities For Improvement:**

1. The final dataset size of 834 samples might be small for training and evaluating large language models. It can limit the generalizability of the findings.
2. The data in the benchmark consists of only Chinese text. It is difficult to assess whether the model's performance pertains specifically to cunning text or to Chinese text in general.
3. There seems to be insufficient distinction between FLUB and general logic problems. I believe a more thorough analysis of the data could be beneficial.
4. While the paper mentions ethical considerations in data cleaning, it would benefit from a more thorough discussion on the ethical implications of using data from forums known for potentially offensive content, even after filtering.

**Relation To Prior Work:**

Yes.

**Summary And Contributions:**

The paper introduces the FaLlacy Understanding Benchmark (FLUB) to assess the fallacy comprehension abilities of LLMs. FLUB consists of tricky, humorous, and misleading texts from a Chinese online forum, Ruozhiba. The benchmark includes three tasks: answer selection, cunning type classification, and fallacy explanation. Experiments reveal that LLMs struggle with fallacy detection and reasoning, which highlights the need for improvement.

---

> ### Author Response · Authors · 2024-08-14
> **Response to Reviewer SvrS**
>
> We appreciate your insightful comments and we hope that our responses will clarify your concerns:
>
> > About the dataset size.
>
> - As discussed in L109-L114, first of all, the reason why FLUB data samples are small is that it comes from the real world, and at the same time, we have extremely strict control over data quality. Therefore, we believe that **the high quality and authenticity of our data can ensure the challenge of evaluating LLMs.** Likewise, other successful previous benchmarks such as TruthfulQA and FreshQA have shown that **data size is not directly related to the evaluation effectiveness of an LLM benchmark.** Finally, we also hope that our work can successfully attract more researchers to participate in the study of fallacy understanding, thereby attracting more people to contribute more data resources to us. As mentioned in the section of "D.7 Maintenance" in the Supplementary Material , **we promise to continue to update and refine our FLUB dataset, and welcome any form of errata and discussion from other researchers in the community. We believe that FLUB will become larger and larger with the joint efforts of the community.**
>
> > About the dataset language.
>
> - Yes, currently FLUB only supports the evaluation of Chinese LLMs. **It is undeniable that Chinese is already the language that cannot be ignored in the NLP community, and due to the cultural differences between different languages, benchmarks specifically used to evaluate Chinese LLMs are also very necessary.** Additionally, we also hope that our work can attract more language researchers to work on the study of fallacy understanding in LLMs. **Therefore, we are also actively working hard to promote the construction of the multilingual version of FLUB.**
>
> > About the distinction between FLUB and general logic problems.
>
> - After careful observation and analysis of the data, we believe that the main distinction between FLUB and general logic problems are as follows:
>   - **The focus is different:** FLUB focuses on LLMs' fallacy understanding ability, which is mainly related to LLMs' perception and intuition of language; while general logic problems focus on LLMs' logical ability, which is mainly related to LLMs' reasoning and thinking.
>   - **The language characteristics are different:** Usually, general logic problems are aimed at the common language used in daily life, while FLUB is aimed at the Internet language used by netizens to surf the Internet.
>
> > About the Ethical Consideration.
>
> - Thank you very much for the reminder. Striving to make our work ethical has always been the bottom line we follow when conducting research. **We understand your concerns and have reviewed our institutional requirements and NeurIPS policies. Our data collection and sharing practices comply fully with these policies**, and we have indeed spent a lot of time and money to ensure that FLUB does not contain any potentially offensive content on the Internet, as mentioned in L75-L76 and L298-L304.

---

### Official Review · Reviewer_2iuj · 2024-07-20
**Review of Paper 930**

**Rating:** 8
**Confidence:** 4
**Correctness:** Yes.
**Clarity:** Yes.

**Review:**

This work is interesting and significant for the research of fallacy understanding ability of LLMs. Generally speaking, the data preprocessing is rigorous. The tasks designed are also reasonable. This benchmark shows interesting results about the ability of different LLMs which may help to improve the ability of common sense understanding in LLMs. However, it is not very clear how GPT-4 score is used in this benchmark. In table 2, the reason why the human results should be multiplied by 2 is not explained. The details should be check about the GPT-4 score part. What's more, 0.69 seems not to be a high score in correlation coefficient. Please make it solid about this part.

**Strengths:**

The topic this paper considering is indeed important.
The way of generating datasets is novel with a rigorous processing pipeline.
The 3 tasks designed help evaluate the ability LLMs in different angles.
The experiments are extensive and have rather interesting discussions.

**Additional Feedback:**

No.

**Documentation:**

Yes.

**Ethics:**

No.

**Limitations:**

The use of GPT-4 score should be more careful.

**Opportunities For Improvement:**

In section 3.4, the part of GPT-4 scores design seems hard to follow. Please make it more reasonable with a good explanation about Table 2, "Note that we multiply the human results by 2 to normalize their range to be the same as the GPT-4 results’ range".
As is pointed out in the Limitations section, it will be a nice try with more examples in English given with the help of professional interpreters.

**Relation To Prior Work:**

Yes.

**Summary And Contributions:**

This paper proposes  a FaLlacy Understanding Benchmark (FLUB) based on data from Ruozhiba (retard forum) which contains cunning texts that are easy for humans to understand but difficult for models to grasp.
Three different tasks are designed to evaluate the fallacy understanding ability of LLMs. Some interesting results are given.

---

> ### Author Response · Authors · 2024-08-14
> **Response to Reviewer 2iuj**
>
> We are very grateful for your recognition and appreciation of our work, and we hope that our reply can solve your confusion.
>
> > About the explanation about Table 2.
>
> - When designing the GPT-4 scoring range and the human scoring range, we have different motivations. **We hope that GPT-4's scoring range can be as unbiased and detailed as possible, so we set its scoring range to 1-10. But this scoring range is too fine-grained and difficult for humans, so we set the human scoring range to 1-5.** Therefore, for comparability of GPT-4 scores and human scores in Table 2, we multiply human scores by 2 to match the range of GPT-4 scores. It is worth emphasizing that we also conducted experiments to observe the scoring situation of GPT-4 when the scoring range was 1-5 and 1-10 respectively. **Experimental results show that the trend of GPT-4 scores does not change with the change of the scoring range, so the normalization operation of multiplying human scores by 2 is reasonable.**
> - We will add the above clarification and make it more reasonable with a good explanation about Table 2 in the final version.
>
> > About the dataset language.
>
> - We strongly agree with you and hope that our focus in this work can be generalized to the broader language community. However, considering that language fallacies themselves are closely related to cultural background, we also hope that our work can attract more researchers to pay attention to and participate in the study of LLMs' fallacy understanding ability. As mentioned in the section of "D.7 Maintenance" in the Supplementary Material , **we promise to continue to update and refine our FLUB dataset, and welcome any form of errata and discussion from other researchers in the community.** We are also actively working hard to promote the construction of the multilingual version of FLUB.

---

### Official Review · Reviewer_VwUX · 2024-08-02
**Well-motivated task and good contribution, but needs more clarification on cunning/fallacy type annotation & classification**

**Rating:** 7
**Confidence:** 4
**Clarity:** It is generally clear, but some parts…

**Review:**

Pros:
- The paper clearly explains the dataset construction method, and the motivation is strong. The type of fallacy/cunning text that is the central theme of this data has not been extensively covered by previous work.
- It is demonstrated that LLMs still struggle with these tasks, with the highest overall performance showing a considerable gap compared to the human baseline.

Cons:
- Although the dataset construction method is generally clear, some details could be elaborated further. I am particularly concerned with the cunning/fallacy classification task. Given the very low performance in this task, it is unclear whether this is due to the weak capabilities of the LLMs or the potentially ambiguous way the fallacy types are defined. Consequently, this raises doubts about the data quality for this specific task. The human baseline for this task (Table 1) is also relatively low (63.69 F1 score), which suggests a lot of disagreements between the human baseline and the gold label.
- The usage of GPT-4 as a judge can potentially result in bias. Some research, such as Panickssery et al. (2024), has shown that LLMs can favor their own generation. From Table 1, it is evident that the GPT-4 score for the GPT-4's output is very high, especially in the CoT setting (9.21 out of 10). Additionally, in Table 2, the correlation between human and GPT-4 evaluations varies depending on the LLM being evaluated, showing the brittleness of this metric.

_*) Panickssery, A., Bowman, S.R. and Feng, S., 2024. Llm evaluators recognize and favor their own generations._

**Strengths:**

- The paper presents a well-motivated task theme, providing a clever way to test both language and commonsense understanding.
- Despite the data being sourced from online forums, which can contain a lot of noise, the authors have carefully curated the relevant posts to ensure data quality.
- The task settings complement each other well, especially the answer selection and fallacy explanation tasks, allowing for a broader analysis of the LLMs' capabilities.

**Additional Feedback:**

Additional question: for the GPT-4 score, why is the scoring range 1-10, while for humans, it is 1-5? I understand that a 1-10 range might be difficult for humans to apply consistently when creating rubrics. If so, why not use a 1-5 range for the GPT-4 score as well to maintain consistency?

**Correctness:**

Overall, the motivation and the dataset construction are sound and clear, except for the fallacy/cunning type annotation, as I mentioned above.

Consequently, some claims in the results section concerning fallacy/cunning type classification need to be double-checked. For example, L209-L212 needs further clarification. Is the observed trend due to the task itself being challenging, or because the types themselves are ambiguous?

**Documentation:**

Yes.

**Ethics:**

I have a minor concern regarding the data copyright/license.

In L69-70, it is stated that the data from Baidu Tieba can be used for "academic research free of charge," which I assume restricts commercial purposes. However, in the supplementary material (L1017-1019), it is stated that FLUB is distributed under the CC BY 4.0 license, which permits commercial use. Should the license be CC BY-NC 4.0 (Creative Commons Non-Commercial) instead? I think this needs to be double-checked by the authors who are more familiar with the Baidu Bar/Baidu Tieba agreement.

**Limitations:**

I think the current limitations section needs to be refined. It should explicitly clarify the limitations in the dataset's usage due to its focus on the cultural and language characteristics specific to Chinese. For example, if the dataset needs to be extended to other languages, careful consideration is required, as direct machine translation might not be appropriate.

Some points also need to be discussed in the limitations section:
- It needs to be explicitly stated that the current automatic metric for fallacy explanation (GPT-4 Score) is still limited and might not reflect the true performance of the LLMs in this task. Readers should interpret the results carefully.
- The multiple-choice task setting might overestimate the current performance of the LLMs, giving an inflated view of their true capabilities. Additionally, there could be ordering option bias in multiple-choice settings, which could also affect the results.

**Opportunities For Improvement:**

Elaborating on the points I mentioned in Cons,
- For the cunning/fallacy types, it is necessary to clarify why and how the 8 types were selected. Some types can be subjective or ambiguous. For example, `Lame Jokes` is potentially subjective. The example of `Factual Error` in Figure 2 ("Which one weighs more, a ton of iron or a ton of cotton?") could also be categorized as `Reasoning Error`. Additionally, although each data point is annotated by three annotators with majority voting, the final label is reviewed and modified by a senior annotator (+ it is also still unclear how the "junior" and "senior" annotators differ). This process can potentially introduce bias (subjectivity). The number of modifications should be clarified, along with the reasons for these modifications (i.e., what kind of annotation errors in the majority voting necessitated changes).
- To reduce bias in the GPT-4 scoring, a more effective approach would be to use an ensemble of several models for scoring, rather than relying solely on GPT-4.

**Relation To Prior Work:**

Yes.

**Summary And Contributions:**

This paper introduces the Fallacy Understanding Benchmark (FLUB) dataset, which contains cunning text with fallacies, such as "Why are there holes in the lotus roots I bought?". These questions are easily understood by humans but are challenging for LLMs to solve. The dataset is provided in Chinese, sourced from online forums, and manually filtered by human annotators.

The dataset includes three tasks:
1. Answer selection (multiple choice format), evaluated by accuracy.
2. Cunning/fallacy type classification, evaluated by F1 score.
3. Fallacy explanation, where LLMs generate explanations for the fallacy texts, evaluated using GPT-4 as a judge, which asked to score the LLMs' output in a range of 1-10.

The authors also benchmark several LLMs using their dataset under various settings, such as w/o CoT and CoT, the influence of in-context learning, and the comparison of human vs. GPT-4 as the evaluator on the fallacy explanation task.

---

> ### Author Response · Authors · 2024-08-14
> **Response to Reviewer VwUX (part 2)**
>
> > About the cunning/fallacy classification task.
>
> - `For the low performance of the cunning type classification task`, we believe that it is not caused by the data quality, but because of the challenge of the task.
>   - **The main reason for the poor performance of the models is that they cannot accurately understand the defined types**. In our previous experiments, we only tell LLMs the Cunning Types, and we think that LLMs might not be able to understand the concept of types just from simple literal words such as `Factual Error` or `Lame Jokes`. Therefore, in our new experiments, we also send the corresponding definitions of various Cunning Types into LLMs. We find that the performance of LLMs has been greatly improved. For example, for GPT-4-Turbo, it has improved from the previous 12.31 became 22.58. Therefore, how to improve the perception ability of LLMs for the cunning types will be the key challenge to improving the performance of LLMs on the cunning/fallacy classification task.
>   - **As for the unsatisfactory performance of humans, after our careful inspection, we believe that the main reason is that the annotators in the human experiments were not carefully trained.** As described in L165-L170, we only introduced the task objective to them, and they had no involvement in the construction process of our dataset, so it is very likely that they did not understand the types we defined very well. The reason why we don't train these annotators carefully is that we want to see the most realistic and primitive performance of humans.
>   - In summary, we believe that the low performance of the cunning type classification task is due to its own challenges, rather than due to data quality issues. We will further clarify related claims, such as L209-L212.
>
> > About the GPT-4 scoring.
>
> - `For the scoring range`, in fact, we first conducted the GPT-4 scoring experiment, and then conducted the human scoring experiment. First of all, we want GPT-4 to provide as unbiased scores as possible, so we set its scoring range to 1-10. When it comes to the human scoring experiment, we find that such a fine-grained range is too difficult for humans. **In our latest experiment, we reset the scoring range of GPT-4 to 1-5, and we found that its trend is basically not fundamentally different from 1-10, and it will not affect the main claims in our paper.** We promise to update the GPT-4 scores in the final version to improve the readability of our paper.
> - `For the potential bias`, we deeply agree with your point of view. Using GPT-4 to evaluate the output of other LLMs is already a widely used method. Although using GPT-4 to evaluate GPT-4 may be biased, using GPT-4 to evaluate other models still has reference value. In addition, not only for the tasks we proposed, but also for other tasks, the community is still actively exploring how to effectively evaluate LLMs. We think that using models to evaluate models is a compromise in its own right. However, we must also emphasize that the focus of our work is to propose new LLMs benchmarks and evaluation tasks. As for designing more effective evaluation metrics, we can leave it to future researchers to continue their work. **We will also refine our Limitations section to remind readers that they should interpret the GPT-4 scores carefully.**
>
> > About the Limitations and Ethical Consideration.
>
> - `For the Limitations section`, we are once again impressed by your insights, and we promise you that we will refine our Limitations section one by one according to your suggestions. **Particularly, we would like to emphasize that our multiple-choice settings should be free of ordering bias since we randomly shuffle the options.**
> - `For the Ethical Consideration`, thank you very much for the reminder. We will change FLUB’s license to CC BY-NC 4.0 (Creative Commons Non-Commercial).

---

> > ### Comment · Reviewer_VwUX · 2024-08-26
> >
> > Thank you for your clarification. Your explanation has addressed my concerns, so I am happy to update my score accordingly. It would be really helpful if you could include your clarifications in the final paper.

---

> > > ### Author Response · Authors · 2024-08-29
> > > **Thank you for your review**
> > >
> > > Thank you very much for your hard work. We promise to add these clarifications in our camera-ready version. We sincerely hope that our work can have a better impact on the research field!

---

> ### Author Response · Authors · 2024-08-14
> **Response to Reviewer VwUX (part 1)**
>
> We appreciate your insightful comments and we hope that our responses will clarify your concerns:
>
> > About the cunning/fallacy classification task.
>
> - `For the definition of 8 cunning types`, in fact, our core authors made a comprehensive summary based on careful observation of the 9,927 initial candidate posts, thus defining 8 types.
>   - **When summarizing the types, we have tried our best to ensure that the various types do not overlap with each other.** As you said, there are still a small number of samples that fall into multiple categories. For these samples, senior annotators and our core authors will discuss carefully and select the main type (i.e., the most obvious type among multiple types) as the annotation result.
>   - In addition, as mentioned in the section of "D.7 Maintenance" in the Supplementary Material, we promise to continue to update and refine our FLUB dataset, and welcome any form of errata and discussion from other researchers in the community. **So if any other researchers find inappropriate annotations, we will be happy to update them in a timely manner.** We believe that a high-quality dataset will continue to evolve with the joint efforts of the community.
>   - We promise to clarify the details about the type definitions more clearly in our final version.
> - `For the annotation of cunning types`, we apologize for troubling you by not providing more details due to space limitations. **In practice, we set up a pre-annotation process (as mentioned in L105-L107) to select senior annotators.**
>   - Specifically, we select senior annotators based on their performance in the pre-annotation process. **If an annotator's annotation success rate is above 95%, he or she will be appointed as a senior annotator. In addition, it is worth mentioning that, all of our formal annotators have a success rate of over 80% in the pre-annotation process.**
>   - At the same time, to avoid bias caused by the subjectivity of annotators as much as possible, our core authors also carefully checked the final annotation results of each data sample.
>   - We promise to add these details of our annotation processes to the final version of our paper.
> - `For the quality of cunning type annotation`, we have sufficient confidence for it.
>   - On the one hand, as described in L124-L128, **we calculate Fleiss’ Kappa to reflect the three junior annotator’s Inter-Annotator Agreement (IAA)**. Our final obtained Fleiss Kappa result is greater than 0.767, which shows that our annotation results have excellent consistency and quality.
>   - On the other hand, **we further ensure annotation quality by checking the annotation and modification results of the senior annotators.** According to our statistics, **senior annotators modified a total of 159 initial annotation results of data samples, that is, the modification rate of senior annotators was 19.06%**. This reflects the excellent workload of our senior annotators and also reflects the high quality of our dataset.
>   - Moreover, after further checking of the modification results of the senior annotators by our core authors, we found that the main reason for the modifications was the disagreement between the senior annotators and the junior annotators on the cunning types (most of the cases were the ones we mentioned before that may fall into multiple types of samples). **For these cases, our core authors made the most reasonable choices and personally modified the annotation results to maximize the quality of the annotation. After all, no one knows the full picture of our work better than our core authors.**
>   - We will clarify the number of modifications , along with the reasons for these modifications in our final version.

---

### Decision · Program_Chairs · 2024-09-26

**Decision:**

Accept (Poster)

**Comment:**

This paper presents a benchmark dataset to test the LLMs ability to answer questions containing fallacies. Humans do naturally well on identifying questions that contain fallacious assumptions, but LLMs are in generally not so good. The problem tackled in this paper and benchmark dataset is interesting, and the dataset is built and explained well. All reviewers agree this paper and the benchmark contain important contributions for the community to test the LLMs' ability to understand fallacies.